# Effect of Surface Roughness on Static Corrosion Behavior of J55 Carbon Steel in CO$_2$-Containing Geothermal Water at 65 °C

**Haitao Bai** [1,*], **Xing Cui** [2], **Rui Wang** [1], **Naixin Lv** [3], **Xupeng Yang** [1], **Ruixuan Li** [1] and **Yun Ma** [1,*]

1 College of Petroleum Engineering, Shaanxi Key Laboratory of Advanced Stimulation Technology for Oil & Gas Reservoirs, Xi'an Shiyou University, Xi'an 710065, China; wangrui_xsyu126@126.com (R.W.); yangxupeng_xsyu126@126.com (X.Y.); liruixuan_xsyu126@126.com (R.L.)
2 College of Chemistry and Chemical Engineering, Key Laboratory of Environment Pollution Control Technology of Oil Gas and Reservoir Protection, Xi'an Shiyou University, Xi'an 710065, China; cuixing@xsyu.edu.cn
3 CNPC Tubular Goods Research Institute, State Key Laboratory of Performance and Structural Safety for Petroleum Tubular Goods and Equipment Materials, Xi'an 710077, China; lvnaixin_xsyu126@126.com
* Correspondence: htbai@xsyu.edu.cn (H.B.); mayun_xian123@126.com (Y.M.); Tel.: +86-150-9186-5583 (H.B.); +86-180-9180-7472 (Y.M.)

**Abstract:** The influence of surface roughness on the static corrosion behavior of J55 carbon steel in CO$_2$-containing geothermal water environment was investigated with respect to average corrosion rate, morphology, chemical composition, corrosion depth, and the cross section of corrosion products. The influence of surface roughness on the CO$_2$ corrosion of J55 carbon steel was then proposed based on the understanding of corrosion at 65 °C. The results show that the static corrosion rate of J55 carbon steel in CO$_2$-containing geothermal water increases with increasing surface roughness. The surface roughness of J55 carbon steel increases 5.3-fold and the CO$_2$ corrosion rate increases by 1.4-fold under different exposure times. The static corrosion rate of J55 carbon steel in CO$_2$-containing geothermal water changes with exposure time. The corrosion rate of J55 carbon steel decreases with the increase in exposure time, and there is little change in the corrosion rate after immersion for 2 days. At the initial stage of corrosion, the corrosion rate of J55 carbon steel was mainly affected by surface roughness. The greater the roughness, the greater the corrosion driving force and the corrosion reaction surface area and therefore the greater the corrosion rate of J55 carbon steel. After immersion for 2 days, a continuous corrosion product layer was formed on the surface of J55 carbon steel and the corrosion rate was mainly affected by the corrosion product layer. The corrosion products of J55 carbon steel are not altered by surface roughness in a CO$_2$-containing geothermal water environment. The corrosion products of J55 carbon steel are FeCO$_3$ and a minute amount of CaCO$_3$.

**Keywords:** surface roughness; CO$_2$ corrosion; J55 carbon steel; geothermal water; corrosion mechanism

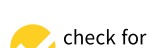



## 1. Introduction

Geothermal energy is a special resource with multiple functions, such as being a heat source and water resource, and its development and utilization value is very high. In recent years, geothermal energy has been used worldwide and has made positive contributions to carbon emission reduction [1–3]. However, the chemical composition of geothermal water is complex, involving dissolved oxygen, Cl$^-$, SO$_4^{2-}$, H$^+$, H$_2$S, CO$_2$, NH$_3$ and other corrosive substances. It also contains Ca$^{2+}$, Mg$^{2+}$, CO$_3^{2-}$, SO$_4^{2-}$ and other scaling substances [4]. The corrosion and scaling of pipeline equipment caused by geothermal water have been hindering the effective utilization of geothermal energy. To ensure the safety and structural integrity of geothermal utilization equipment during operation and maintenance, it is essential to evaluate the failure analysis of such equipment, especially in environments where geothermal water contains CO$_2$ [5,6].

At present, the research on the corrosion caused by geothermal water has mainly focused on pipeline and equipment failure analysis [7,8] and the impact of environmental

factors (temperature, pressure, etc.) [9,10] and little attention has been paid to the impact of the surface properties of pipes. However, the surface microstructure of metallic materials plays an important role in the corrosion reaction of metals, which can lead to differences in physical and chemical properties of the material surface [11,12]. At the same time, the electrochemical properties of metallic substances could cause significant surface effects due to the different surface roughness, which could cause changes in the physical quantities related to the corrosion reaction, thus affecting the progress of the corrosion reaction [13]. So far, however, there has been little discussion regarding the effect of surface roughness on the corrosion behavior of carbon steel in $CO_2$-containing geothermal water environment. However, scholars have also performed preliminary studies on the effect of metal surface states on corrosion reactions in other corrosive media and formed a certain degree of consensus [14–22]. The initial corrosion predominantly starts on the peaks of the sample surface, while the corrosion in the valleys is relatively small [14–16]. Li et al. [17] pointed out that the corrosion rate of metal surface increases with the increase in sample surface roughness. The reason for this is that the peaks of the sample surface are able to provide higher electrochemical activity than the valleys, which allows the metal surface electrons to react more easily with environmental media [18–20]. The research of Pistorius et al. [18] shows that the greater surface roughness of stainless steel, the more pits will be produced after corrosion. Sang et al. [19] found that the smoother the metal surface, the higher its pitting resistance and the less likely it is to form metastable pitting or pits. Lee et al. [20] concluded that the corrosion behavior of steel is closely related to the roughness of the material surface, but when the metal surface roughness is greater than 0.5 μm, its corrosion resistance is no longer related to the surface roughness. Sasaki et al. [21] pointed out that in solution rich in $Cl^-$, the pitting potential of metal samples decreased linearly with the increase in surface roughness. In addition, the research results of Asma et al. [22] suggest that in the solution without a corrosion inhibitor, the corrosion rate of carbon steel and copper increases with the increase in material surface roughness. The current research mainly focuses on the impact of surface roughness on corrosion at the initial stage of corrosion and does not evaluate the impact of metal surface roughness on the long-term service safety of pipelines and equipment. Therefore, it is necessary to study the effect of surface roughness on long-term metal corrosion.

Carbon steel is widely used in the development and application of geothermal energy due to its excellent mechanical properties and low cost [23]. $CO_2$ corrosion and its control have been important topics in the field of oil and gas development [24–26]. The research results of $CO_2$ corrosion in the oil and gas industry can provide a reference for the corrosion of $CO_2$-containing geothermal water. Pertinent research focuses on the impact of environmental and material factors on corrosion behavior [24–26]. For example, the corrosion rate and mode are determined by the corrosive medium and environment [25]. The corrosion rate is affected by temperature and pressure, which alter the properties of the corrosion product layer [27–32]. The $CO_2$ corrosion rate of carbon steel shows a trend of first increasing and then decreasing with the increase in temperature and the maximum value of the corrosion rate also changes accordingly [27–29]. At 100 °C, the maximum corrosion rate of the P110 pipe steel was reported by Li [30]. The $CO_2$ corrosion rate of carbon steel was positively correlated with $CO_2$ partial pressure [31,32]. However, in a 25 wt.% NaCl solution at 65 °C, the corrosion rate of carbon steel (API 5CT L80) was found to vary negligibly with $CO_2$ pressure (4, 8 and 12 MPa), as reported by Choi et al. [33].

The purpose of this article is to investigate the effect of surface roughness on the corrosion behavior of J55 carbon steel under static conditions in $CO_2$-containing geothermal water environments. Specimens with different surface roughness were prepared by grinding with different SiC emery papers. The corrosion rates and the maximum corrosion depth were determined. Additionally, the morphology and composition of the corrosion products layer formed were characterized on metal surfaces to investigate the corrosion mechanism of J55 carbon steel exposed to a $CO_2$-containing geothermal water.

## 2. Materials and Methods

### 2.1. Material

The material used in this work was J55 carbon steel, and the element composition is shown in Table 1. The specimen for the weight loss test was machined into a 50 mm × 10 mm × 3 mm size, with a hole with a bore diameter of 6 mm drilled to enable the suspension of samples in tests solutions, resulting in an exposed area of 13.6 cm$^2$.

**Table 1.** The main element contents of J55 carbon steel.

| Element | C | Si | Mn | P | S | Cr | Ni | Cu | Fe |
|---|---|---|---|---|---|---|---|---|---|
| Concentration (wt.%) | 0.36 | 0.30 | 1.45 | 0.016 | 0.004 | 0.051 | 0.009 | 0.07 | balance |

The experimental medium was geothermal water, which was taken from a geothermal well in Xianyang City, Shaanxi Province, People's Republic of China, and its pH value was 7.64. The main content of geothermal water is shown in Table 2.

**Table 2.** The main contents of the geothermal water.

| Ion | $Cl^-$ | $HCO_3^-$ | $CO_3^{2-}$ | $SO_4^{2-}$ | $Ca^{2+}$ | $Mg^{2+}$ | $Na^+$ |
|---|---|---|---|---|---|---|---|
| Concentration (mg·L$^{-1}$) | 408.40 | 282.33 | 153.47 | 3775.74 | 126.07 | 25.69 | 2105.21 |

### 2.2. Surface Finish

Immediately before placing the specimens into the test solution, the specimens were ground with SiC emery papers, washed in tap water, ultrasonicated in deionized water for 3 min in order to remove any residual SiC grains, degreased with acetone, rinsed with deionized water and dried with an air drier. The schematic diagram of the rough surface preparation process is shown in Figure 1. The specimen was pressed onto SiC emery papers using an index finger and ground in one direction. The surface of the specimen was processed into an oriented, uniform, matte surface. The roughness of the specimen's surface after grinding was controlled by the number of SiC emery paper meshes.

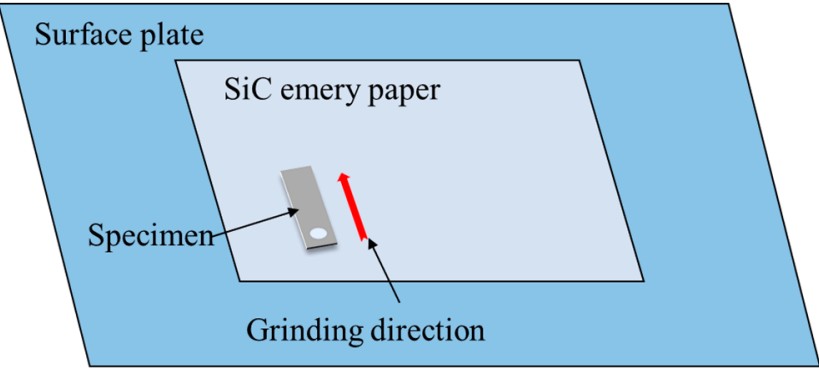

**Figure 1.** Sketch map of rough surface preparation process.

### 2.3. Surface Characterization

The surface roughness was determined through an optical digital microscope (OLYMPUS DSX500, Olympus Corporation, Tokyo, Japan). The roughness of the specimens was evaluated via non-contact optical observation. Ten measurements were recorded on each sample at 1000× magnification. When the OLYMPUS DSX-500 optical digital microscope was used to measure the surface roughness of the specimens, the line roughness *Ra* was extended to three dimensions. The average absolute deviation of the roughness irregularities, *Sa*, was used to quantify surface roughness.

*Sa* is the average absolute deviation of the roughness irregularities from the center surface. For a profile defined by *n* surface height measurements on the evaluation area *A*:

$$Sa = \frac{1}{A} \int_A \int /Z(x,y)/dxdy \tag{1}$$

where *A* is the evaluation area, *Z* is the distance from the measuring point to the central surface, and *x* and *y* are the number of sampling points in *x* and *y* directions in the evaluation area *A*. *Sa* is the arithmetic mean deviation of the area morphology. It was used to characterize the roughness of two-dimensional surface morphology.

The surface area of the measurement area was calculated by multiplying the cross-sectional curve length and the longitudinal length. The section curve was extracted using an optical digital microscope. The cross-sectional curve length and surface hemispherical particle radius were measured using Image-Pro Plus 6.0 software.

$$A = L_1 \times L_2 \tag{2}$$

where *A* is the evaluation area, $\mu m^2$; $L_1$ is the cross-sectional curve length, $\mu m$; and $L_2$ is longitudinal length, $\mu m$.

Equation (3) was used to calculate the area ratio.

$$R_A = \frac{A}{A_0} \tag{3}$$

where $R_A$ is the area ratio; *A* is the surface area of $Sa \neq 0$ $\mu m$, $\mu m^2$; and $A_0$ is surface area of $Sa = 0$ $\mu m$, $\mu m^2$.

### 2.4. Weight-Loss Method

The weight-loss method was used to conduct the corrosion test in a high-pressure autoclave, with the schematic diagram shown in Figure 2. A total of 2 L of geothermal water in a beaker was placed inside the autoclave, and a small amount of nitrogen gas was used to purge the dissolved oxygen in the solution at a pressure of 0.5 MPa [34] and a temperature of 65 °C for 4 h. Then, the autoclave was pressurized with pure $CO_2$ gas to a pressure value of 5 MPa, and $N_2$ gas was pressurized to a total pressure value of 15 MPa. Finally, the test conditions were maintained in a static state for different test times. After the corrosion process, corroded samples were exposed to Clarke solution [35] to remove the corrosion products. Equation (4) was used to calculate the average corrosion rate.

$$r_{corr} = \frac{8.76 \times 10^4 \times (m - m_t)}{S \times t \times \rho} \tag{4}$$

where $r_{corr}$ is average corrosion rate, $mm \cdot year^{-1}$, and *m* is the mass of the sample before the experiment, g. $m_t$ is the mass of the sample after the experiment, g. *S* is the whole surface contacted with solution, $cm^2$. $\rho$ is the density of tested steel, $g/cm^3$, which in the case of carbon steel is 7.86 $g/cm^3$. *t* is the immersion duration, h. The average corrosion rate error of each test was calculated through three parallel specimens.

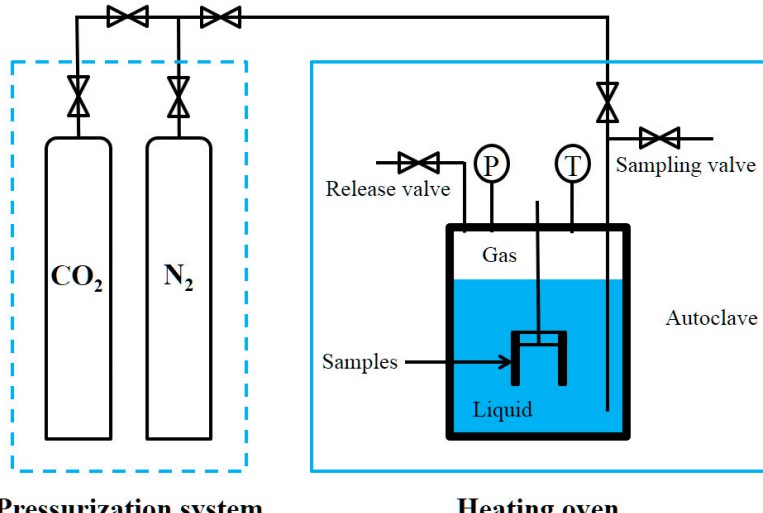

**Figure 2.** Flow chart of steel corrosion rates evaluation system (mass loss method).

### 2.5. Surface Analysis

A scanning electron microscope (FEI Quanta 600 F microscopy, FEI Corporation, Hillsboro, TX, USA) was used for the microstructures analysis of the surface and cross-section of the corroded samples [33]. An energy spectrum analyzer (Oxford INCA energy 350, Oxford Instrument, Oxford, UK) was used for the elemental composition analysis of the corrosion product scales. X-ray diffraction (Bruker D8 XRD, Bruker Corporation, Karlsruhe, Germany) was used for the compositional analysis of the corroded samples. The optical digital microscope (Olympus DSX500, Olympus Corporation, Tokyo, Japan) was used for the maximum corrosion depth of the corroded samples [36].

## 3. Results

### 3.1. Surface Morphology Observation

Figure 3 shows the surface morphology of specimens after being ground with different SiC emery papers. After surface treatment, most areas of the samples surface show irregular undulating morphology and the scratch direction was consistent. The frequency and amplitude of the wave in the sample surface after further SiC emery paper grinding was more than that of the sample surface after 1000# SiC emery paper grinding. The sample with larger roughness were able to accommodate more wave peaks and troughs than the sample with smaller roughness. As shown in Figure 4, the surface roughness of the J55 carbon steel after being gradually ground decreased with SiC emery papers with increasing mesh numbers.

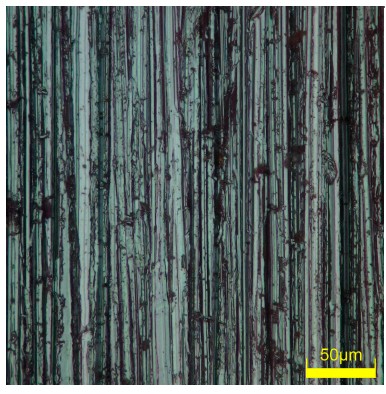 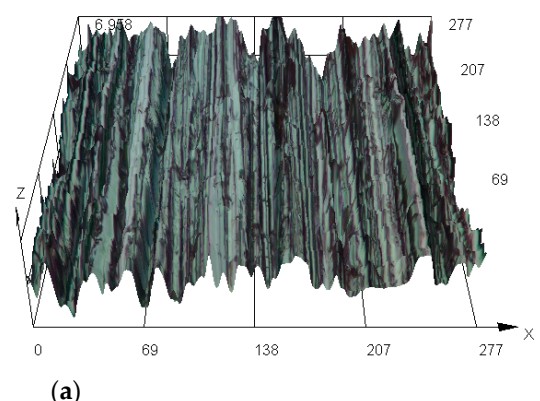

(a)

**Figure 3.** *Cont.*

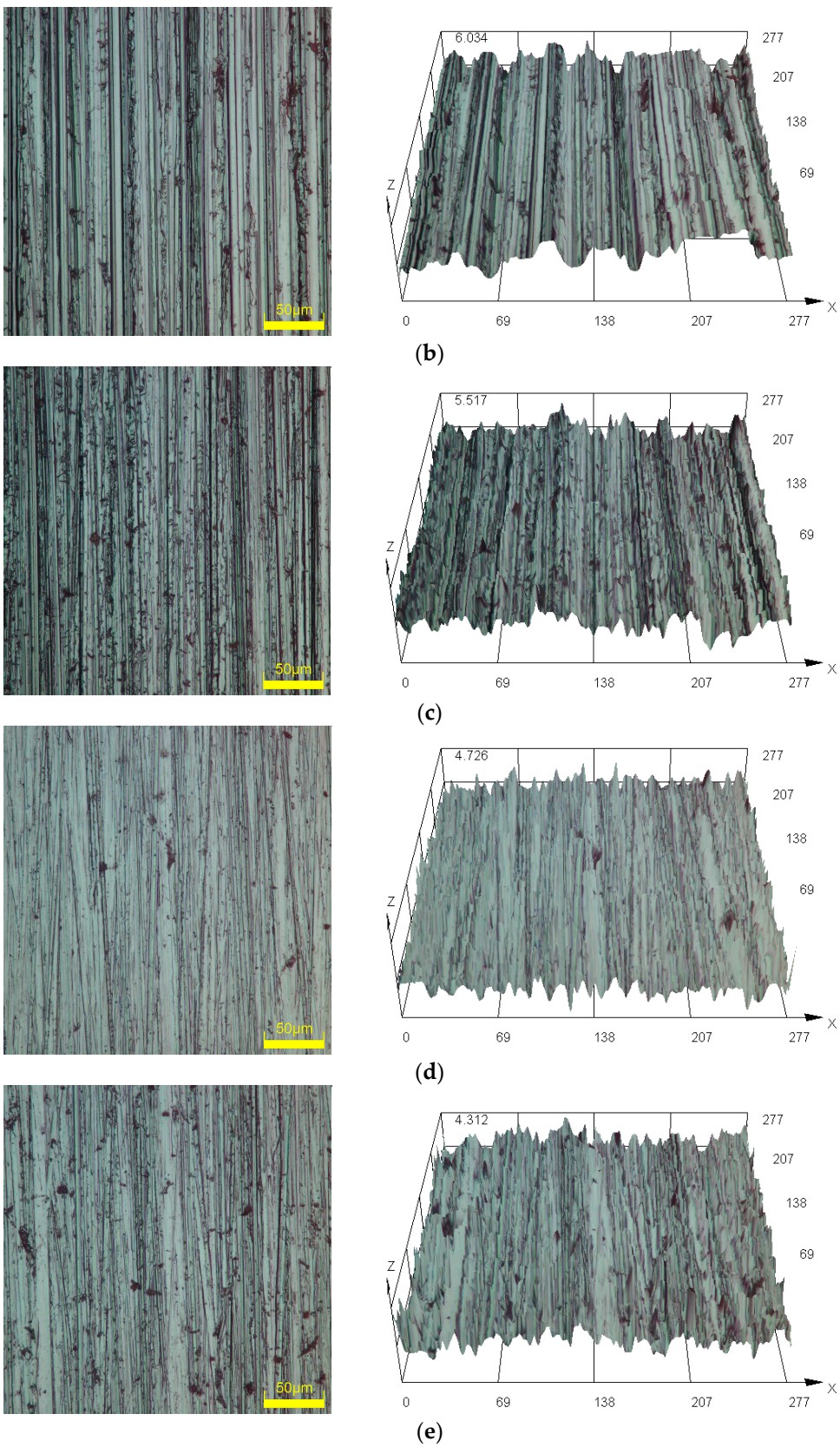

**Figure 3.** Surface morphology of specimens after grinding with different SiC emery papers: (**a**) 100# SiC emery paper; (**b**) 180# SiC emery paper; (**c**) 320# SiC emery paper; (**d**) 400# SiC emery paper; (**e**) 1000# SiC emery paper.

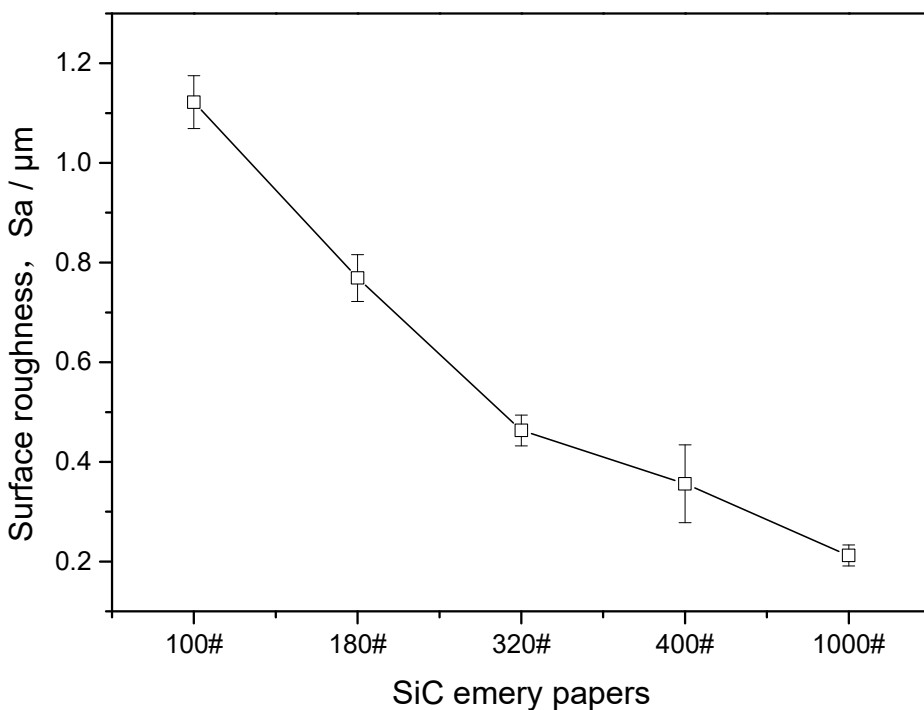

**Figure 4.** Surface roughness of specimens after grinding with different SiC emery papers.

*3.2. Weight Loss Tests*

The average corrosion rates of J55 carbon steels at $P_{CO_2}$ = 5 MPa and 65 °C after different exposure times are provided in Figure 5. It can be observed that the exposure time and surface roughness have a significant impact on the corrosion of J55 carbon steel. The corrosion rate changes little after the immersion time exceeds 48 h, which is in good agreement with the literature [33]. The effect of exposure time on the average corrosion rate is linked to the formation of $FeCO_3$ layer. With an increase in exposure time, the $FeCO_3$ layer formed on the surface of J55 carbon steel will also increase and thicken. The corrosion product layer hinders the further corrosion of J55 carbon steel, and the corrosion rate will decrease along with exposure time, eventually reaching a constant value. It can also be seen that the corrosion rate increased with the increase in the surface roughness. Significant scratches and gullies were present on the surface of samples with large surface roughness and height differences between peaks and valleys. The corrosion product layer could be formed at the peak, but due to the tip effect, the corrosion product layer could easily break down and form corrosion microcells, thus increasing corrosion [37]. The greater the roughness of the samples in solution, the stronger the corrosion sensitivity; the lower the surface finish, the larger the increase in specific surface area, surface defects and impurities; and the higher the corrosion tendency in the solution, the more likely the occurrence of corrosion [38]. The lower the surface roughness, the higher the finish of the samples surface, which is conducive to the formation of a dense oxide layer on the surface, so as to protect the matrix, slow down the corrosion rate and improve the corrosion resistance of the samples [39]. As shown in the results of the weight loss tests, the effect of surface roughness on the corrosion rate decreases with the increase in immersion time.

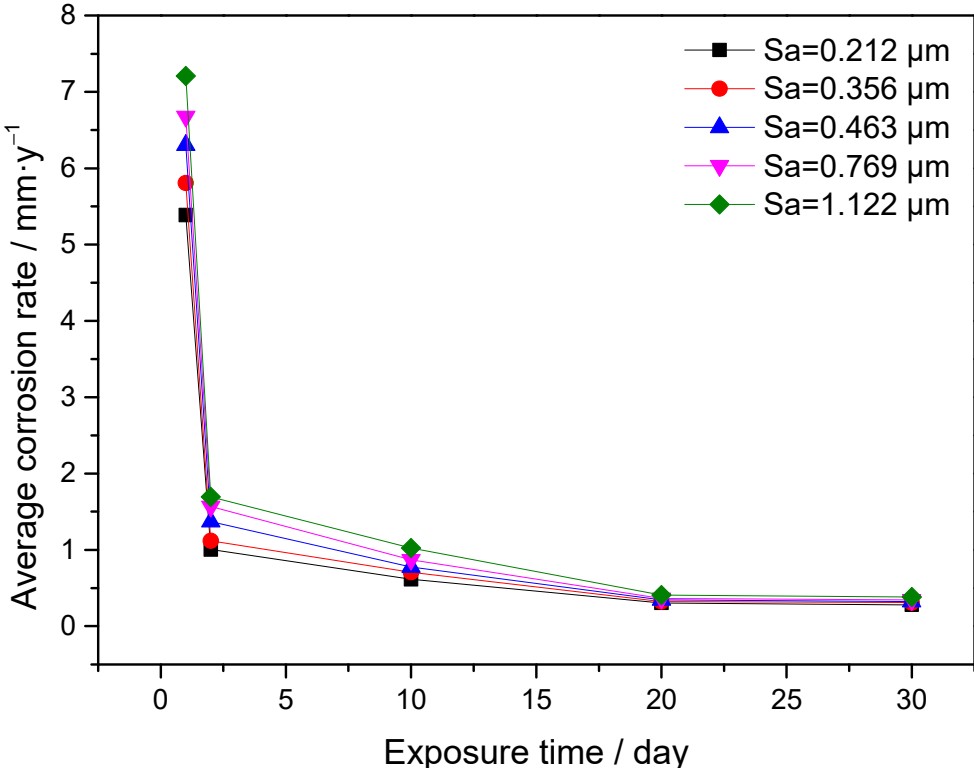

**Figure 5.** Average corrosion rate determined by mass loss technique as a function of exposure time for J55 carbon steel immersed in $CO_2$-containing geothermal water.

### 3.3. Characterization of Corrosion Scales Developed on the Surface

Figure 6 shows the SEM images of corrosion scale morphology formed on the steel surface ($Sa$ = 0.463 μm) after exposure to $CO_2$-containing geothermal water at 65 °C and static conditions. With the increase in exposure time, the corrosion products gradually covered the surface of the sample. After 1 day of exposure, a small number of corrosion products were produced on the surface of the sample but did not completely cover the surface (Figure 6a). After 2 days of exposure, the corrosion products completely covered the surface of the sample and protected the carbon steel substrate (Figure 6b). As the corrosion time further increased, the corrosion product layer became denser (Figure 6c,d). The corrosion products gradually increase with the increase in corrosion time, the corrosion products gradually become denser and the protection ability is enhanced, which can explain the change in the trend of corrosion rate in Figure 5 well.

Figures 7–11 shows the SEM images of corrosion scale morphology formed on J55 steel surface as a function of surface roughness in $CO_2$-containing geothermal water, at the same magnifications (×100 or ×2000). EDS was performed on the corrosion product scales of the tested samples. Table 3 shows the EDS spectra of the corrosion scale in the inner surface of the area outlined in blue in Figures 7–11. It can be seen that the corrosion products completely cover the surface of J55 carbon steel, and no polishing trace on the surface can be observed. The $FeCO_3$ crystal and fine cracks can be clearly observed from the images magnified by 2000. The corrosion products of J55 carbon steel surface decrease with the increase in surface roughness. A large number of pits were found in the SEM image of the corrosion scale morphology formed on the steel surface in $CO_2$-containing geothermal water at $Sa$ = 1.122 μm condition.

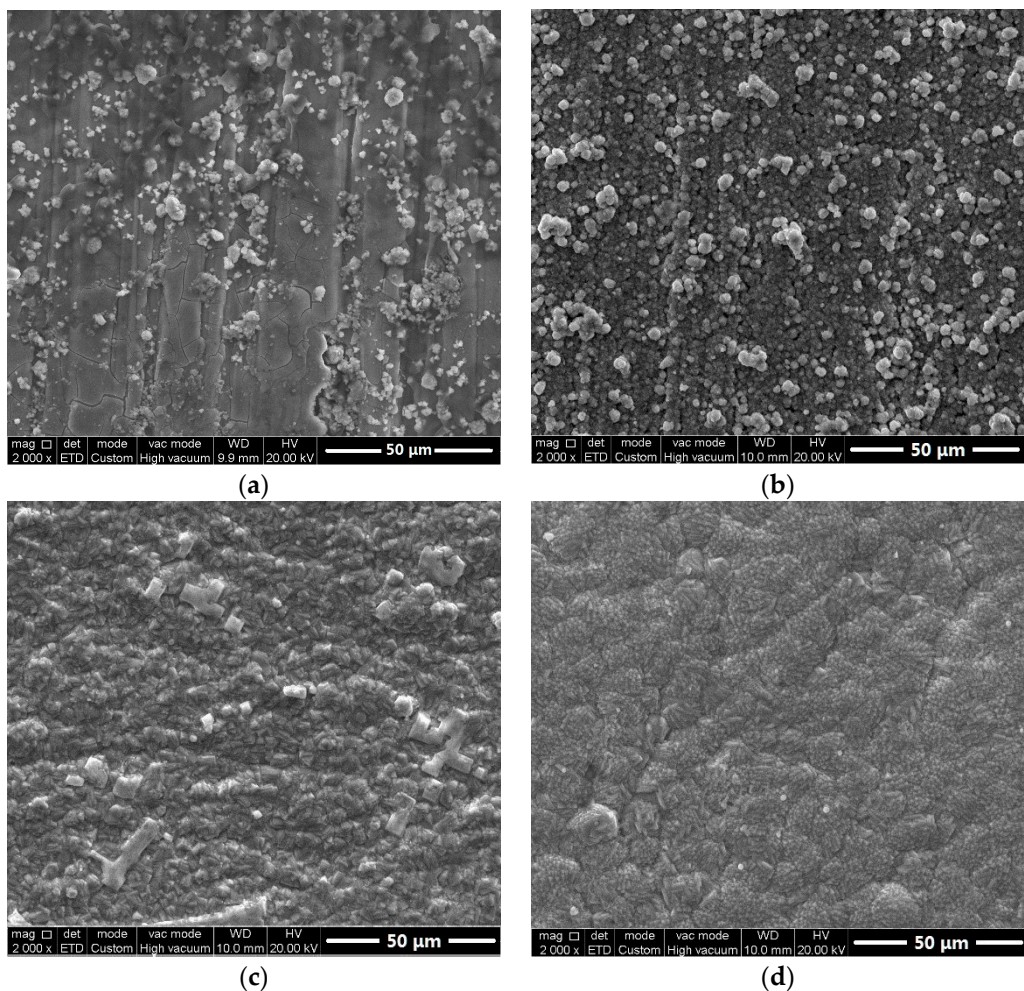

**Figure 6.** SEM images of corrosion scale morphology formed on steel surface ($Sa$ = 0.463 μm) after exposure to $CO_2$-containing geothermal water at 65 °C and static conditions: (**a**) after exposure for 1 day, ×2000; (**b**) after exposure for 2 days, ×2000; (**c**) after exposure for 10 days, ×2000; and (**d**) after exposure for 20 days, ×2000.

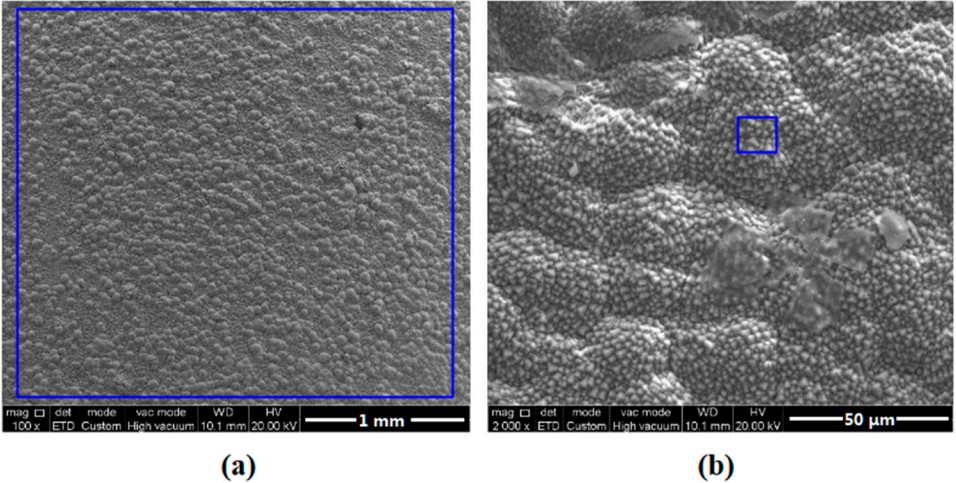

**Figure 7.** SEM images of corrosion scale morphology formed on steel surface ($Sa$ = 0.212 μm) in $CO_2$-containing geothermal water after exposure for 30 days at 65 °C and under static conditions: (**a**) ×100; (**b**) ×2000.

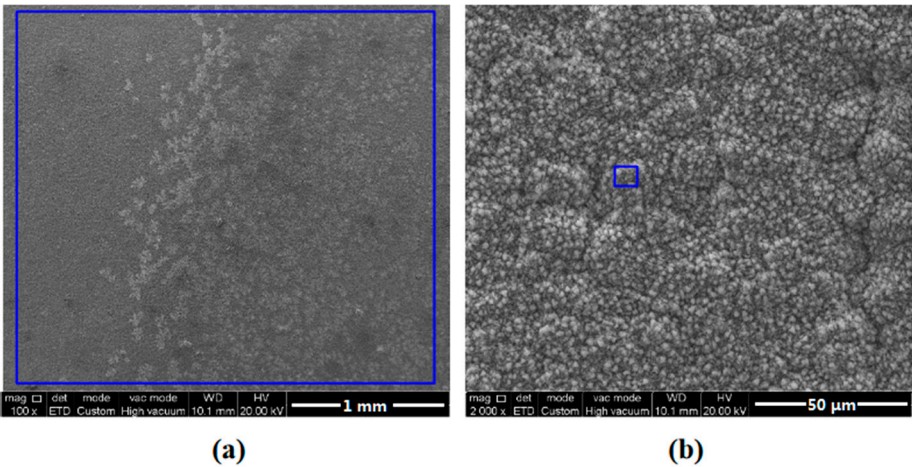

**(a)**                                                  **(b)**

**Figure 8.** SEM images of corrosion scale morphology formed on steel surface ($Sa$ = 0.356 μm) in $CO_2$-containing geothermal water after exposure for 30 days at 65 °C and under static conditions: (**a**) ×100; (**b**) ×2000.

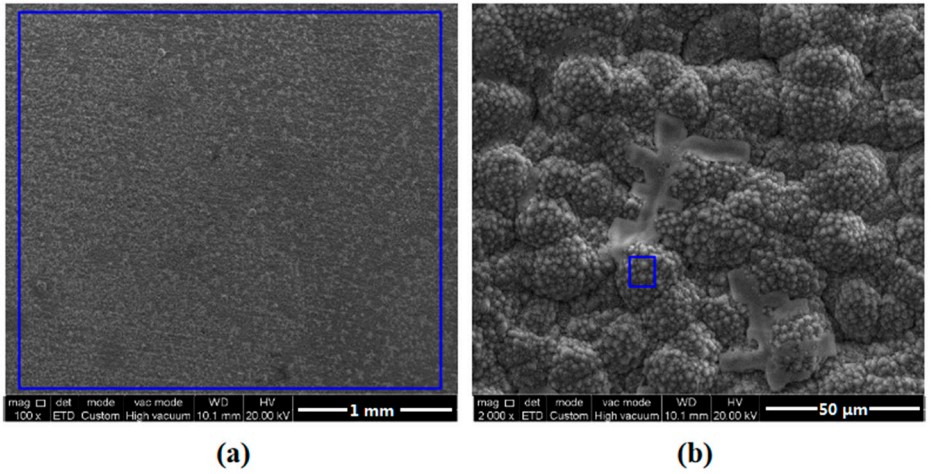

**(a)**                                                  **(b)**

**Figure 9.** SEM images of corrosion scale morphology formed on steel surface ($Sa$ = 0.463 μm) in $CO_2$-containing geothermal water after exposure for 30 days at 65 °C and under static conditions: (**a**) ×100; (**b**) ×2000.

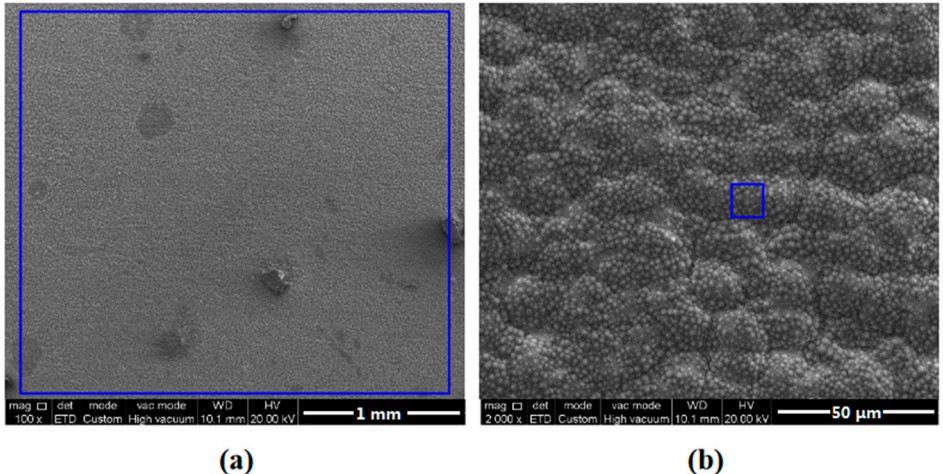

**(a)**                                                  **(b)**

**Figure 10.** SEM images of corrosion scale morphology formed on steel surface ($Sa$ = 0.769 μm) in $CO_2$-containing geothermal water after exposure for 30 days at 65 °C and under static conditions: (**a**) ×100; (**b**) ×2000.

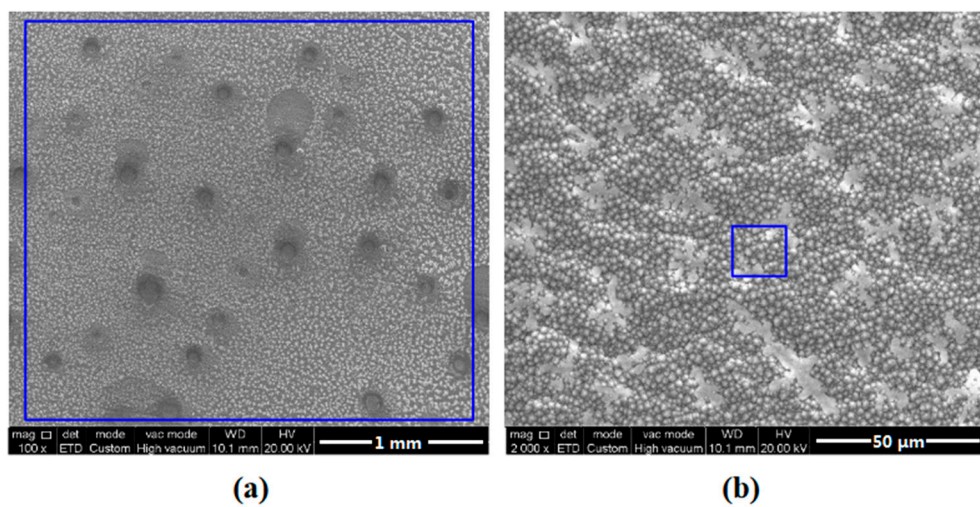

**Figure 11.** SEM images of corrosion scale morphology formed on steel surface (*Sa* = 0.1.122 μm) in CO$_2$-containing geothermal water after exposure for 30 days at 65 °C and under static conditions: (**a**) ×100; (**b**) ×2000.

**Table 3.** EDS analysis of the corrosion scale after immersion in CO$_2$-containing geothermal water.

| Elements (At%) | *Sa* = 0.212 μm | | *Sa* = 0.356 μm | | *Sa* = 0.463 μm | | *Sa* = 0.769 μm | | *Sa* = 1.122 μm | |
|---|---|---|---|---|---|---|---|---|---|---|
| | Whole | Local | Whole | Local | Whole | Local | Whole | Local | Whole | Local |
| C K | 20.78 | 21.13 | 23.84 | 15.29 | 19.52 | 17.56 | 20.00 | 24.85 | 20.96 | 18.04 |
| O K | 48.47 | 51.13 | 45.84 | 55.02 | 46.11 | 51.38 | 49.26 | 39.34 | 48.89 | 50.46 |
| Cl K | 1.61 | 0.53 | 0.55 | / | 0.45 | 0.21 | 1.19 | 1.61 | 1.57 | / |
| Ca K | 2.33 | 6.08 | 2.64 | 6.54 | 2.09 | 1.99 | 2.00 | 4.42 | 2.30 | 2.13 |
| Mn K | / | / | / | / | / | / | 0.46 | 0.75 | 0.58 | 0.30 |
| Fe K | 26.81 | 21.13 | 27.13 | 23.15 | 31.83 | 28.86 | 27.09 | 29.03 | 25.70 | 29.07 |
| total | 100.0 | 100.0 | 100.0 | 100.0 | 100.0 | 100.0 | 100.0 | 100.0 | 100.0 | 100.0 |

The large square region outlined in blue was measured at a magnification of 100, with a broad measurement area, and the measurement results are able to characterize the elemental composition of all the corrosion products. The small square area outlined in blue was measured at a magnification of 2000, with a small measurement area, and the measurement results are able to characterize the elemental composition of local corrosion products. In geothermal water containing CO$_2$, the primary elements of the corrosion products are Fe, O and C, with a ratio of approximately 1:3:1. This indicates that the corrosion products are primarily composed of FeCO$_3$. At the same time, the corrosion products contain a small amount of Ca, so there was mixed carbonate (Fe$_x$Ca$_{1-x}$CO$_3$) present in the corrosion products [40,41]. The Cl element was also noted in the corrosion products, which might lead to pitting corrosion. At *Sa* = 0.769 and 1.122 μm, a minor constituent of alloying elements from carbon steel, Mn, was detected.

Figure 12 illustrates the XRD patterns of the surface layer on corroded samples that were immersed in CO$_2$-containing geothermal water. The composition of the corrosion product layer in CO$_2$-containing geothermal water was similar and mainly consisted of the complex salt of CaCO$_3$ and FeCO$_3$. The observed result can be attributed to the isomorphous substitution of metal cations during CO$_2$ corrosion [42]. When the [Fe$^{2+}$] × [CO$_3{}^{2-}$] in the medium exceeded FeCO$_3$ solubility product K$_{sp}$ (FeCO$_3$), the FeCO$_3$ was deposited on the

metal surface. The replacement of $Fe^{2+}$ in the $FeCO_3$ crystal by $Ca^{2+}$ in the solution and the formation of the $Fe(Ca)CO_3$ complex can be expressed as:

$$Ca^{2+} + FeCO_3 \text{ (s)} \rightarrow Fe^{2+} + CaCO_3 \text{ (s)} \tag{5}$$

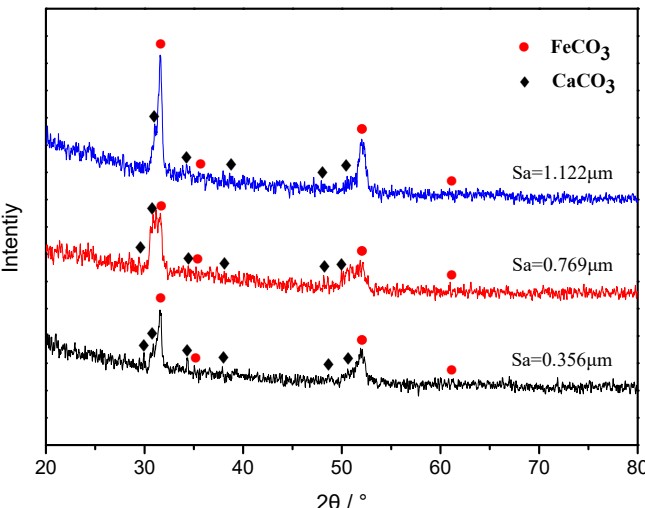

**Figure 12.** XRD patterns of surface layer on the corroded samples.

### 3.4. Maximum Corrosion Depth Tests

Figure 13 shows the relationship between the maximum corrosion depth and the ratio between the pitting corrosion rate and the average corrosion rate of the J55 carbon steel surface, as well as surface roughness after removing the corrosion product layer using an acidic solution in $CO_2$-containing geothermal water. The maximum corrosion depth increases with the increase in surface roughness, which may be attributed to the influence of initial surface conditions. The trend of the ratio between pitting corrosion rate and average corrosion rate on the surface of J55 carbon steel is first decreased and then increasing. When $Sa = 0.769$ μm and $Sa = 1.122$ μm, the pitting rate/average corrosion rate ratios were higher than 4, indicating that local corrosion occurred on the surface of the carbon steel [42]. At $Sa = 1.122$ μm, the corrosion depth was the largest at 162.817 μm, which corresponded to 1.9809 mm·year$^{-1}$. This penetration rate was considerably greater than the weight-loss corrosion rate (0.3835 mm·year$^{-1}$) shown in Figure 5, thereby confirming a local attack.

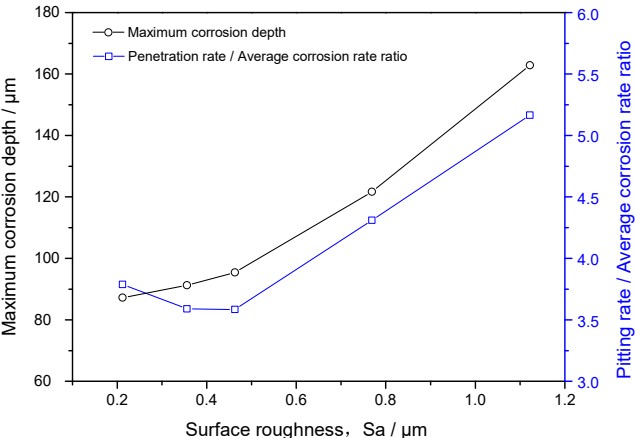

**Figure 13.** Maximum corrosion depth and pitting rate/average corrosion rate ratio of the J55 carbon steel surface as a function of surface roughness for steel immersed in $CO_2$-containing geothermal water.

## 4. Discussions

### 4.1. Influence of Surface Effect

In the system of hemispherical protruding particles on rough surfaces, each surface has a corresponding surface tension, namely, the surface Gibbs function. For this hemispherical particle, the Gibbs function relationship between the particle radius and the electrochemical reaction can be derived [43,44]:

$$\Delta_r G_m = \Delta_r G_m^b + \Delta_r G_m^s \tag{6}$$

where $\Delta_r G_m$ is the molar Gibbs function of an electrochemical corrosion reaction; $\Delta_r G_m^b$ is the internal Gibbs function of the mole (when the surface is smooth) and $\Delta_r G_m^s$ is the Gibbs function of the molar surface.

Of these:

$$\Delta_r G_m^b = \Sigma v_B \mu_B^b \tag{7}$$

$$\Delta_r G_m^B = \Sigma v_B \mu_B^S = \Sigma 2 v_B \sigma_B M_B / \rho_B r_B \tag{8}$$

where $v_B$ is the stoichiometric number of component $B$ in the chemical reaction equation; $\mu_B^b$ is the degree of internalization of substance $B$; $\mu_B^S$ is the surface degree of substance $B$; $\sigma_B$ is the surface tension of substance $B$; $M_B$ is the molar mass of substance $B$; $\rho_B$ is the density of substance $B$; and $r_B$ is the surface hemispherical particle radius.

If the electrochemical reaction of the metal corrosion is designed as a reversible primary electrode cell, then the total Gibbs function of the metal corrosion reaction on the rough surface and the relationship between the total electric potential is:

$$\Delta_r G_m = \Delta_r G_m^b + \Delta_r G_m^s = -nE^b F + (-nE^s F) \tag{9}$$

$$E = E^b + E^s = E^b - \Sigma 2 v_B \sigma_B M_B / nF \rho_B r_B \tag{10}$$

where $E$ is the total driving force of the rough surface, $V$; $E^b$ is surface driving force, $V$; and $E^S$ is the internal driving force, $V$.

In the same corrosion reaction process $v_B$, $\sigma_B$, $M_B$, $\rho_B$, $n$ and $F$ are constant, then the smaller $-\sum 1/r_B$, the larger the total electric potential $E$ of the surface and the more likely the surface is to corrode. The radius of the curvature of all small projecting hemispherical particles of a certain cross-section of the rough surface of J55 carbon steel after grinding with different sandpaper in Figure 3 was measured and calculated as $-\sum 1/r_B$, as shown in Figure 14. In the electrochemical corrosion process, the steel is the anode (substance $B$), that is, the reactant ($v_B < 0$), and Figure 14 shows that the greater the roughness, the smaller $-\sum 1/r_B$, and if the total surface electric potential $E$ is higher, the more likely corrosion is to occur. A rough surface can increase the driving force of corrosion cells. When the surface roughness increases, a strong surface effect can be produced. Therefore, the rate of chemical corrosion reaction and thermodynamic parameters is changed and the corrosion rate is further increased. On the rough surface of the metal, if only rough electrochemical corrosion exists, the electrode potential of the protruding particles is low, and as the anode, it is continuously corroded and dissolved. The electrode potential of the concave hole is high, and the corrosion product is continuously deposited as the cathode.

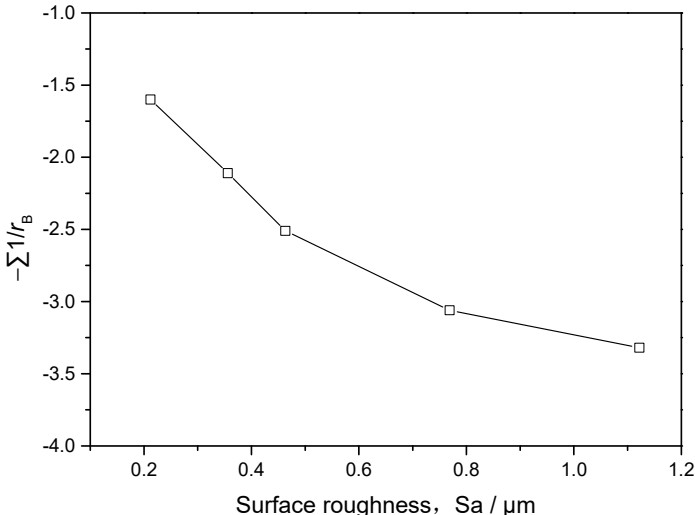

**Figure 14.** $-\sum 1/r_B$ of J55 carbon steel surface as a function of surface roughness.

### 4.2. Effect of Corrosion Product Layer

The general view on the electrochemical process of $CO_2$ corrosion is that $CO_2$ dissolves in water to form $H_2CO_3$ and the $H_2CO_3$ in solution reacts with Fe to cause corrosion [45,46]. The main reaction equations are as follows:

$$CO_2 + H_2O \rightarrow H_2CO_3 \tag{11}$$

$$H_2CO_3 \rightarrow H^+ + HCO_3^- \tag{12}$$

$$HCO_3^- \rightarrow H^+ + CO_3^{2-} \tag{13}$$

$$2H^+ + Fe \rightarrow Fe^{2+} + H_2 \tag{14}$$

$$Fe^{2+} + CO_3^{2-} \rightarrow FeCO_3 \tag{15}$$

Total corrosion reaction:

$$CO_2 + H_2O + Fe \rightarrow FeCO_3 + H_2 \tag{16}$$

At this time, $Fe^{2+}$ on the sample surface and corrosion reaction products form a series of insoluble ferrite precipitates on the sample surface, which impede further corrosion reactions.

According to the reaction equation, the formation of corrosion products is related to the diffusion of $Fe^{2+}$. According to Fick's first diffusion law, the diffusion process of water-soluble ion $Fe^{2+}$ generated by corrosion is related to the diffusion rate, and the equation is shown in Equation (17).

$$\frac{dn}{dc} = -DA\frac{dc}{dx} \tag{17}$$

where $dn/dc$ is material diffusion rate; $D$ is diffusion coefficient; $A$ is diffusion channel surface area; and $dc/dx$ is concentration gradient.

At the initial stage of corrosion reaction, few corrosion products were generated on the surface, diffusion coefficient $D$ was a constant value. The diffusion rate of $Fe^{2+}$ was the same as the surface area $A$ and the concentration gradient $dc/dx$ was dependent. Figure 15 shows the section curve of specimens with different surface roughness. When the surface roughness of the sample increases, the curve of the sample surface fluctuates more, the wave crest is sharper and the curve length of the sample surface is larger; thus, the

concentration gradient is larger and the curvature radius is larger. The section curve length was measured by Image-Pro Plus 6.0 software, and the area ratio of the different roughness surfaces in Figure 3 was calculated, as shown in Table 4. It can be seen from Table 4 that the cross-section curves of samples with different surface roughness vary greatly, the actual surface area of the sample with $Sa = 0.212$ μm was 1.52-fold the actual surface area of the sample with $Sa = 0$ μm and the difference between the actual surface area and the theoretical surface area gradually increases as the surface roughness increases, while the diffusion channel surface area $A$ becomes larger. Therefore, the surface of a sample with significant levels of roughness is more prone to corrosion.

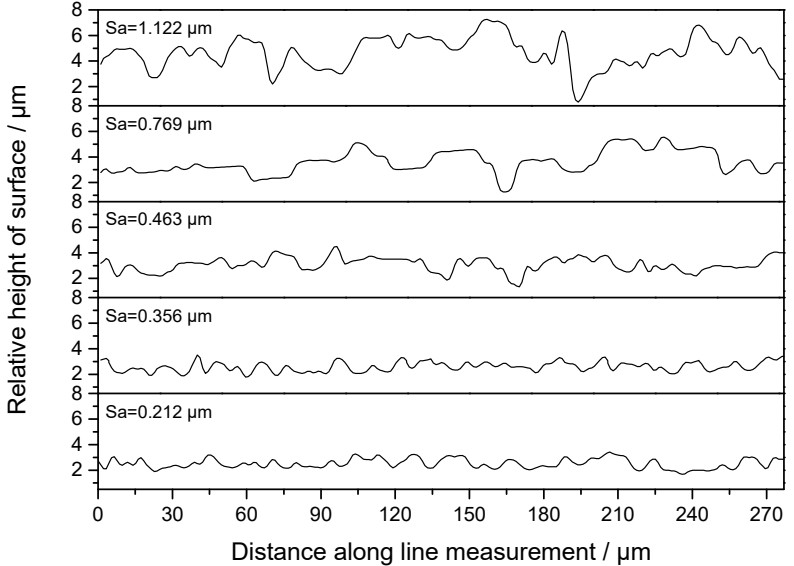

**Figure 15.** The section curve of specimens with different surface roughness.

**Table 4.** The section curve length and surface area ratio of specimens with different surface roughness.

| $Sa$ (μm) | 0 | 0.212 | 0.356 | 0.463 | 0.769 | 1.122 |
|---|---|---|---|---|---|---|
| Length (μm) | 277 | 422.26 | 467.64 | 500.91 | 594.20 | 621.42 |
| $R_A$ | 1.00 | 1.52 | 1.69 | 1.81 | 2.15 | 2.24 |

The studies [14,19] concluded that early corrosion mainly began on the peaks of the specimen's surface, whereas corrosion occurred in the troughs in relatively few cases. With the development of corrosion behavior, the corrosion reaction gradually shifted from the peak areas to the trough areas, and corrosion reaction on the surface of the specimen gradually converged on the same. With the extension of the immersion time, corrosion products gradually covered the surface of J55 carbon steel, and corrosion product layer affected the diffusion coefficient $D$. Figure 16 shows the cross-sectional SEM images of the corrosion layer of J55 carbon steel after corrosion under different conditions. It can be seen that the thickness of the corrosion product layer with the increase in surface roughness, the increasing trend and corrosion rate measurement results are consistent. The corrosion product layer is comprehensive and dense and has strong protective abilities toward the substrate. Therefore, when exposed for 30 days, the corrosion rate of carbon steel in $CO_2$-containing geothermal water is low. At the same time, pits were found on the substrate at $Sa = 1.112$ μm.

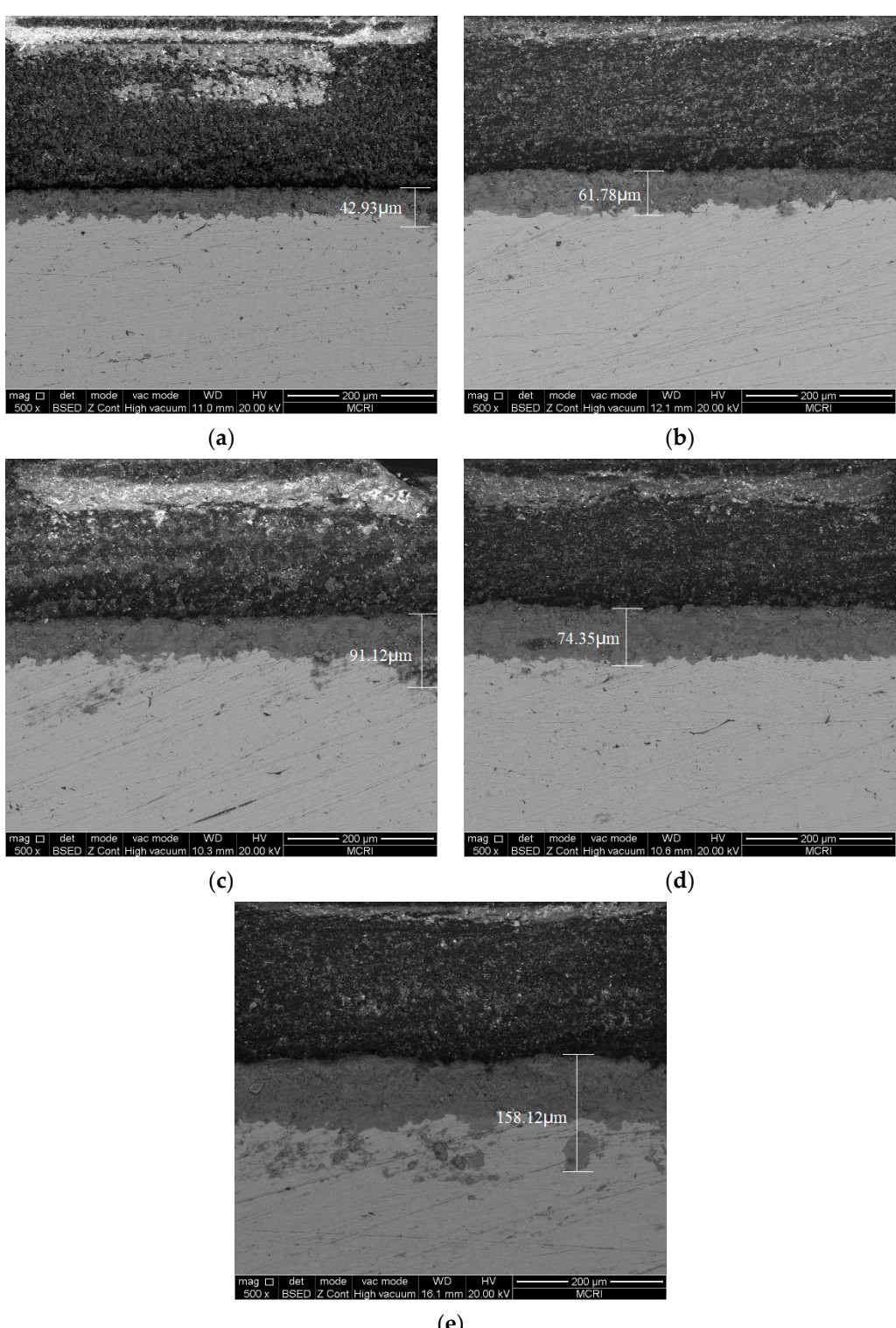

**Figure 16.** Cross-sectional SEM images of the corrosion scales on J55 carbon steel in $CO_2$-containing geothermal water after exposure for 30 days at 65 °C and under static conditions: (**a**) $Sa$ = 0.212 μm; (**b**) $Sa$ = 0.356 μm; (**c**) $Sa$ = 0.463 μm; (**d**) $Sa$ = 0.769 μm; and (**e**) $Sa$ = 1.112 μm.

In addition, plastic deformation improves metal surface activity and promotes corrosion reactions. The microstructure residual strain on the metal surface is increased, which provides more electrochemically active sites and produces mechanical–chemical interactions, thus promoting the corrosion of the metal [47].

### 4.3. Corrosion Mechanism

The corrosion rate of J55 carbon steel in $CO_2$-containing geothermal water is influenced by surface roughness and the corrosion product layer, as confirmed by the above-presented results. In order to clarify the influence mechanism of surface roughness on the $CO_2$ corrosion of J55 carbon steel, corrosion models are proposed, as shown in Figure 17. The two stages describing the formation of corrosion are as follows:

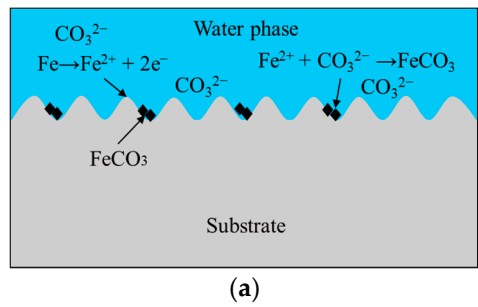 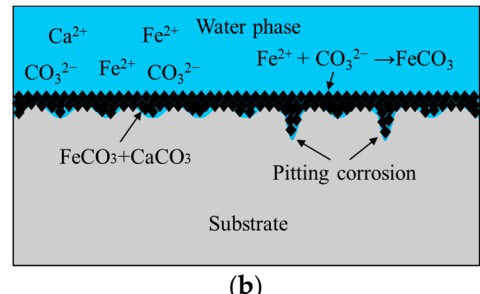

(**a**)                                                                           (**b**)

**Figure 17.** Schematic models for the corrosion behavior of J55 carbon steel with different surface roughness in $CO_2$-containing geothermal water at 65 °C and under static conditions: (**a**) at the initial corrosion (exposure time of less than 2 d); (**b**) after the formation of complete corrosion product layer (exposure time of more than 2 d).

Stage 1 shown in Figure 17a: At the initial stage of corrosion (exposure time of less than 2 days), J55 carbon steel surface is exposed to $CO_2$-containing geothermal water. On the rough surface of J55 carbon steel, the protruding areas are continuously corroded and dissolved as anodes. The electrode potential at the concave hole is high, becoming a cathode, and the corrosion products are continuously deposited. Rough surface can increase the driving force of corrosion cells and the area of the corrosion reaction, as shown in Figure 14 and Table 4. Therefore, at the initial stage of corrosion, the surface roughness significantly affects the corrosion rate of J55 carbon steel in $CO_2$-containing geothermal water, and the corrosion rate increases with the increase in roughness. At the initial stage of corrosion, the potential of $E_m$, at which point the metastable pit or pits start to grow on the surface, depends on surface roughness. The rougher the surface, the lower the $E_m$ values, indicating that the metastable pit or pits can begin to grow on the surface more easily [20].

Stage 2 shown in Figure 17b: After the formation of complete corrosion product layer (exposure time of more than 2 days), the corrosion reaction is related to the diffusion rate of $CO_2$-containing geothermal water and $Fe^{2+}$ in the corrosion product layer. The density and thickness of the corrosion product layer are the main factors that affect the diffusion coefficient $D$ and the corrosion rate. Under experimental conditions, the corrosion product layer of J55 carbon steel produced in $CO_2$-containing geothermal water is dense and thick, which restricted the transport of $H^+$ and $Fe^{2+}$ in and out, respectively. Therefore, it is able to protect the surface of carbon steel and reduce the corrosion rate. Under $Sa$ = 1.112 μm conditions, the corrosion product layer covers the pits formed during the initial stage of corrosion, forming a "occluded cell", which intensifies the pitting rate [48].

### 5. Conclusions

Based on the static corrosion behavior of J55 carbon steel in $CO_2$-containing geothermal water environment at 65 °C, the following can be concluded:

(1)    At 65 °C, the static corrosion rate of J55 carbon steel in $CO_2$-containing geothermal water increases with increasing surface roughness. The surface roughness of J55 carbon steel increases by 5.3-fold and the $CO_2$ corrosion rate increases by 1.4-fold under different exposure times.

(2)    The static corrosion rate of J55 carbon steel in $CO_2$-containing geothermal water changes with exposure time, and there is little change in the corrosion rate after immersion for 2 days.

(3)　After immersion for 2 days, a complete corrosion product layer was formed on the surface of J55 carbon steel, and the corrosion rate was mainly affected by the corrosion product layer. The corrosion rate of J55 carbon steel displays little change.

(4)　In $CO_2$-containing geothermal water environment, the surface of J55 carbon steel was covered with $FeCO_3$ and a minute amount of $CaCO_3$.

(5)　At the initial stages of corrosion, the surface roughness affects the corrosion rate through the corrosion driving force and the corrosion reaction surface area. After the formation of complete corrosion product layer, the corrosion product layer is the primary factor affecting the corrosion rate.

**Author Contributions:** H.B., X.C., R.W. and R.L. conceived and designed the experiments; H.B. and X.Y. analyzed the data and wrote the paper; H.B., Y.M. and N.L. proposed the corrosion model. All authors have read and agreed to the published version of the manuscript.

**Funding:** This research was funded by the Natural Science Basic Research Program of Shaanxi, grant number 2021JQ-600, and the Scientific Research Program Funded by Shaanxi Provincial Education Department, grant number 21JK0841.

**Institutional Review Board Statement:** Not applicable.

**Informed Consent Statement:** Not applicable.

**Data Availability Statement:** Not applicable.

**Conflicts of Interest:** The authors declare that they have no conflict of interest.

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
