# Peer review of "Effect of Surface Roughness on Static Corrosion Behavior of J55 Carbon Steel in CO2-Containing Geothermal Water at 65 °C"

_coatings, doi:10.3390/coatings13050821_

Round 1

Reviewer 1 Report

The authors invistigatrd the  Effect of surface roughness on static corrosion behavior of J55 carbon steel in CO2 containing geothermal water at 65 ℃. The research findings are  intersting and  manuscript well written. However, some of the following questions needed to be improve the quality of the paper.

1. Specify the some of the important properties of geothermal water and how it influence on corrosion 

2. How to control the surface roughness during sand paper process (Scratching of J55 carbon steel)

3. Please highlight the direction of sand paper scratching on surface in the sem images

4. In Fig. 15, please highlight the depth of corrosion in sem images 

5. Conclusions should be specific, please update the conclusions 

Author Response

An overview of the revision

Thanks for your suggestions and advices on our paper. We have revised the manuscript according to your detailed suggestions. Words in red are the changes we have made in the text. We have revised the problems and discussed your arguments as follows.

Comment 1: Specify the some of the important properties of geothermal water and how it influence on corrosion.

Response 1: The content of dissolved oxygen, CO2, and H2S in geothermal water are important factors that affect corrosion. This article focuses on the static corrosion of CO2 containing geothermal water, so CO2 corrosion was described in lines 81-98.

Comment 2: How to control the surface roughness during sand paper process (Scratching of J55 carbon steel).

Response 2: We have added to the text to make the experimental approach easier to understand, please see lines 122-127 and Figure 1.

Comment 3: Please highlight the direction of sand paper scratching on surface in the sem images.

Response 3: The surface of the test piece was completely covered by corrosion products, so the direction of sand paper scratching cannot be highlighted.

Comment 4: In Fig. 15, please highlight the depth of corrosion in sem images.

Response 4: We have made changes, please see Fig. 16.

Comment 5: Conclusions should be specific, please update the conclusions.

Response 5: We have updated the conclusions, please see lines 459-474.

Reviewer 2 Report

REVIEWER COMMENTS ON__Effect of surface roughness on static corrosion behavior of J55 carbon steel in CO2 containing geothermal water at 65℃.

General Comments

The subject matter is interesting but the manuscript is replete with many errors.

The abstract is poorly written. The abstract needs to be written using appropriate terms and provide some quantitative data

The manuscript contains a lot of factually inaccurate statements and mix-ups. In the introduction, the authors should have a paragraph highlighting the grades of material used in piping geothermal water in order to extract energy from it.  

Secondly, the authors did not highlight the importance of their work.

Thirdly the authors failed to describe at all or in sufficient details in the methods section how some of the measurements were made which would help a reader either replicate the work or draw his/her own conclusions.

Fourthly, The authors should include a schematic showing exactly how the steel samples were polished or ground to obtain the surface finishes they presented.

Fifthly, arguments and conclusions were drawn without sufficient facts. The authors based the discussion on the electrochemical nature of corrosion without any electrochemical data presented, not even the open circuit potentials (OCP)  variations of the various surfaces with time.

Sixth, their discussion on mechanism was based on potential of pitting or metastable pits but it has no support from the data presented. It is suggested that authors carry out “cyclic potentiodynamic polarization scans” which can provide information on differences in pitting behaviour and pitting potential with surface roughness.

Other errors detected have been highlighted under specific comments below often with suggestions that would help the authors remedy these errors.

Specific Comments

LINE 14: Write “with respect to” instead of “about”.

LINE 16: Delete “mechanism”,

LINE 18: Write “corrosion products” instead of corrosion.

LINE 18-19: Write “initially” instead of “At the initial of corrosion,”

LINE 18-21: The 2 sentences here lack clarity. Please rephrase.

LINE 20-21: “The surface roughness affects the corrosion rate through the corrosion cell electromotive force and the corrosion reaction surface area.”

This sentence is lacking in clarity. Rephrase using appropriate terms.

What do the authors mean by “corrosion cell electromotive force”?

LINE 30-31: “Geothermal energy is a special resource with multiple functions such as heat source, water resources, and mineral resources, and its development and utilization value is very high.”

This sentence as presented is not factual and is confusing. Geothermal energy cannot be water resources and mineral resources ??

Please rephrase and present your ideas in a factually accurate manner and with clarity,

LINE 33-35: Citations are needed to support the quoted constituent of geothermal water in the 2 sentences.

LINE 36-37: “The corrosion and scaling of geothermal water have been hindering the efficient utilization of geothermal energy.”

This is another factually inaccurate sentence. Geothermal water does not undergo corrosion or scaling. It is the pipes that are used to channel and exploit its energy potential that suffer corrosion and scaling. Please rephrase.

LINE 41: “the corrosion of geothermal water…..” is not correct. Geothermal water does not corrode! Rephrase to achieve factual accuracy.

LINE 42: What do the authors mean by “field failure analysis”? This is not clear.

LINE 44: Write “metallic materials” instead of “metal materials”.

LINE 47: “metal solid substances”? The phrase does not make sense.  Rephrase to improve clarity.

LINE 51-53: ”However, scholars have also made a preliminary study on the effect of metal surface state on corrosion reaction in other corrosive media, and formed a certain degree of consensus recognition results.”

Citation(s) is/are needed to support this statement.

The statement provokes these questions (to which there are no answers in the manuscript nor pointers to sources for answers); Which scholars made the mentioned studies? In which other corrosive media did these scholars study?

LINE 54 and 55: “wave peak positions” and “wave valley positions” are inappropriate terms. Rephrase the statement using appropriate terms.

LINE 56: Delete “reaction”.

LINE 57-59: Citations are needed to support this statement, explanation/claim.

LINE 60: Which metal? Mention the metal their findings relate to.

LINE 63: “believes” is an inappropriate word in this context. Rephrase.

LINE 63: Delete “metal”.

LINE 66: Write “solution” instead of “the solution”.

LINE 71: Write “does not” instead of “cannot”

LINE 72-73: Delete “in order”.

LINE 74-75: Citations are needed to support this statement.

LINE 76: Write “media” instead of “medium”.

LINE 76-77: Citations are needed to support this statement.

LINE 86: Write “reported that the corrosion rate” instead of “the corrosion rate”.

LINE 93: Delete “mass”.

LINE 95: Write “corrosion products layer formed” instead of “formed corrosion products layer”.

LINE 97: Write “on metal surfaces” instead of “by surface analysis”.

LINE 103-104: What is the meaning of “φ = 6 mm”? Express it clearly.

LINE 106: Write “was” instead of is?

LINE 117-130: Authors should include a sentence with citations of previous works that have used such methods as used in this work to evaluate surface roughness.

LINE 133: Delete “A”. Write “was placed inside” instead of “was added to”.

LINE 133: Was the 2L of geothermal water in a beaker? conical flask? Etc.?. Please indicate this in the sentence.

LINE 136: “pressured”?  Not the appropriate word, please rephrase using a more appropriate word(s).

LINE 137-138: Rephrase the sentence to read like an action instead of a statement.

LINE 142: Corrosion rate does not have units of “mm/ a”.

LINE 142: m and mt are not qualities of the metal sheet. Use appropriate scientific terma and avoid such ambiguity.

LINE 146: Rephrase “by three parallel specimens of each test”.

LINE 154: Write “evaluated by” instead of “performed with”.

LINE 155-156: Write “determined” instead of “analyzed”.

LINE 161: Write “ground” instead of “grounded”.

LINE 161-169: How was the grinding operation carried out to get such surfaces? This must be clearly stated in the work.

Did you polishing in one direction? With only one Sic paper grade? In a rotating disc? Progressively with finer SiC papers until the target grade?

As you can see the lack of this information creates ambiguity in the mind of the reader and makes it difficult to appreciate the surface morphology data presented.

LINE 168: Write “gradually ground” instead of (grounded gradually”.

LINE 180: Write “grinding” instead of “grounded”.

LINE 184: Write “grinding” instead of “being grounded”.

LINE 186: Write “PCO2” instead of “Pco2

LINE 180-191: It is necessary to present data on both the surface profiles after 48 hours and the chemical composition of the surface after 48 hours to determine why the corrosion rates convergence after 48 hours occur irrespective of the initial surface roughness.

LINE 205: Write “shown” instead of “showed”.

LINE 222: Write “the change in the” instead of “the change”.

LINE 227-230: Write “day” or “days” as appropriate instead of “d”.

LINE 239: Delete “pitting”.

LINE 244: Write “days” instead of “d”.

LINE 247: Write “days” instead of “d”.

LINE 251: Write “days” instead of “d”.

LINE 254: Write “days” instead of “d”.

LINE 257: Write “days” instead of “d”.

LINE 265: Specify the alloying elements detected.

LINE 266: Write “scale after” instead of “scale of”.

LINE 267: “XRD spectra” is incorrect. Use appropriate description/words.

LINE 271-273: Rephrase this sentence to improve clarity.

LINE 276: “XRD spectra” is incorrect. Use appropriate description/words.

LINE 278-279: What is the significance of this ratio “penetration rate/average corrosion rate ratio”? What does it measure or indicate? Highlight it in the work.

LINE 284: Delete “was”.

LINE 297-299: The meaning of this sentence is difficult to grasp. Rephrase to improve express your idea with clarity or delete the sentence.

LINE 320-321: What justifies these distinctions of E?

LINE 324-326: How was this measured? This was not indicated in the methods section.

LINE 328-336: This argument is unclear.

LINE 340: Sentence lacks clarity. Rephrase.

LINE 340-342: Rephrase this sentence

LINE 346: Write “equations are as follows:” instead of “formula is as following:”.

LINE 348: Write “form” instead of “formed”.

LINE 360: “dependent” on what? The sentence is not clear.

LINE 365-366: How were the area ratios calculated? Again, this information is not presented in the materials and methods section.

LINE 374: How consistent are the profiles presented in Figure 14 across the sample surface to permit the calculation of surface areas?

LINE 392: Write “days” instead of “d”.

LINE 405: Write “days” instead of “d”.

LINE 409-415: This arrangement is not sufficiently supported by data presented. To talk of pitting and pitting potential as mentioned in Lines 412-415 electrochemical test data must be included that shows the pitting behaviour as the surface roughness changes. For this, a cyclic potentiodynamic polarization can be sufficient.

LINE 439-441: Rephrase this sentence to improve clarity.

Author Response

An overview of the revision

Thanks for your suggestion and advice on our paper. We have revised the manuscript according to your detailed suggestions. We have carefully checked and improved the English writing in the revised manuscript. Words in red are the changes we have made in the text. We have revised the problems and discussed your arguments as follows.

Comment 1: The abstract is poorly written. The abstract needs to be written using appropriate terms and provide some quantitative data.

Response 1: It has been changed, please see lines 13-30.

Comment 2: The manuscript contains a lot of factually inaccurate statements and mix-ups. In the introduction, the authors should have a paragraph highlighting the grades of material used in piping geothermal water in order to extract energy from it. 

Response 2: It has been added, please see lines 79-80.

Comment 3: Secondly, the authors did not highlight the importance of their work.

Response 3: It has been changed, please see lines 75-78.

Comment 4: Thirdly the authors failed to describe at all or in sufficient details in the methods section how some of the measurements were made which would help a reader either replicate the work or draw his/her own conclusions.

Response 4: It has been changed, please see lines 118-181.

Comment 5: Fourthly, The authors should include a schematic showing exactly how the steel samples were polished or ground to obtain the surface finishes they presented.

Response 5: We have added to the text to make the experimental approach easier to understand, please see lines 122-127 and Figure 1.

Comment 6: Fifthly, arguments and conclusions were drawn without sufficient facts. The authors based the discussion on the electrochemical nature of corrosion without any electrochemical data presented, not even the open circuit potentials (OCP)  variations of the various surfaces with time.

Response 6: Thanks for your suggestion. Electrochemical testing can provide direct evidence, but we believe that measurement and theoretical calculation of surface properties can also reflect the impact of surface roughness on corrosion electrochemistry.

Comment 7: Sixth, their discussion on mechanism was based on potential of pitting or metastable pits but it has no support from the data presented. It is suggested that authors carry out “cyclic potentiodynamic polarization scans” which can provide information on differences in pitting behaviour and pitting potential with surface roughness.

Response 7: Thanks for your suggestion. Our laboratory does not currently have the conditions for high pressure electrochemical testing, and your suggestion will be our research direction.

Comment 8: Other errors detected have been highlighted under specific comments below often with suggestions that would help the authors remedy these errors.

Response 8: It has been changed.

Specific Response

LINE 14: Write “with respect to” instead of “about”.

It has been changed, please see lines 14.

LINE 16: Delete “mechanism”,

It has been changed, please see lines 16.

LINE 18: Write “corrosion products” instead of corrosion.

It has been changed.

LINE 18-19: Write “initially” instead of “At the initial of corrosion,”

It has been changed, please see lines 23.

LINE 18-21: The 2 sentences here lack clarity. Please rephrase.

It has been changed, please see lines 23-28.

LINE 20-21: “The surface roughness affects the corrosion rate through the corrosion cell electromotive force and the corrosion reaction surface area.”

This sentence is lacking in clarity. Rephrase using appropriate terms.

What do the authors mean by “corrosion cell electromotive force”?

It has been changed, please see lines 24-26. “corrosion cell electromotive force” is wrong and should be“corrosion electromotive force”.

LINE 30-31: “Geothermal energy is a special resource with multiple functions such as heat source, water resources, and mineral resources, and its development and utilization value is very high.”

This sentence as presented is not factual and is confusing. Geothermal energy cannot be water resources and mineral resources ??

Please rephrase and present your ideas in a factually accurate manner and with clarity,

It has been changed, please see lines 35-36.

LINE 33-35: Citations are needed to support the quoted constituent of geothermal water in the 2 sentences.

It has been changed, please see line 40 and line 489.

LINE 36-37: “The corrosion and scaling of geothermal water have been hindering the efficient utilization of geothermal energy.”

This is another factually inaccurate sentence. Geothermal water does not undergo corrosion or scaling. It is the pipes that are used to channel and exploit its energy potential that suffer corrosion and scaling. Please rephrase.

It has been changed, please see lines 40-42.

LINE 41: “the corrosion of geothermal water…..” is not correct. Geothermal water does not corrode! Rephrase to achieve factual accuracy.

It has been changed, please see lines 46-47.

LINE 42: What do the authors mean by “field failure analysis”? This is not clear.

It has been changed, please see lines 46-47.

LINE 44: Write “metallic materials” instead of “metal materials”.

It has been changed, please see lines 49-50.

LINE 47: “metal solid substances”? The phrase does not make sense.  Rephrase to improve clarity.

It has been changed, please see line 52.

LINE 51-53: ”However, scholars have also made a preliminary study on the effect of metal surface state on corrosion reaction in other corrosive media, and formed a certain degree of consensus recognition results.”

Citation(s) is/are needed to support this statement.

The statement provokes these questions (to which there are no answers in the manuscript nor pointers to sources for answers); Which scholars made the mentioned studies? In which other corrosive media did these scholars study?

It has been changed, please see line 51.

LINE 54 and 55: “wave peak positions” and “wave valley positions” are inappropriate terms. Rephrase the statement using appropriate terms.

It has been changed, please see line 60.

LINE 56: Delete “reaction”.

It has been changed, please see line 61.

LINE 57-59: Citations are needed to support this statement, explanation/claim.

It has been changed, please see lines 62-64.

LINE 60: Which metal? Mention the metal their findings relate to.

It has been changed, please see line 65.

LINE 63: “believes” is an inappropriate word in this context. Rephrase.

It has been changed, please see line 68.

LINE 63: Delete “metal”.

It has been changed, please see line 68.

LINE 66: Write “solution” instead of “the solution”.

It has been changed, please see line 71.

LINE 71: Write “does not” instead of “cannot”

It has been changed, please see line 76.

LINE 72-73: Delete “in order”.

It has been changed, please see line 78.

LINE 74-75: Citations are needed to support this statement.

It has been changed, please see lines 80-81.

LINE 76: Write “media” instead of “medium”.

It has been changed, please see line 85.

LINE 76-77: Citations are needed to support this statement.

It has been changed, please see line 86.

LINE 86: Write “reported that the corrosion rate” instead of “the corrosion rate”.

It has been changed, please see lines 86-87.

LINE 93: Delete “mass”.

It has been changed, please see line 102.

LINE 95: Write “corrosion products layer formed” instead of “formed corrosion products layer”.

It has been changed, please see line 104.

LINE 97: Write “on metal surfaces” instead of “by surface analysis”.

It has been changed, please see line 106.

LINE 103-104: What is the meaning of “φ = 6 mm”? Express it clearly.

It has been changed, please see line 113.

LINE 106: Write “was” instead of is?

It has been changed, please see line 115.

LINE 117-130: Authors should include a sentence with citations of previous works that have used such methods as used in this work to evaluate surface roughness.

This is a very good suggestion, but the equipment used in previous works is different from that in this study.

LINE 133: Delete “A”. Write “was placed inside” instead of “was added to”.

It has been changed, please see line 156.

LINE 133: Was the 2L of geothermal water in a beaker? conical flask? Etc.?. Please indicate this in the sentence.

It has been changed, please see line 156.

LINE 136: “pressured”?  Not the appropriate word, please rephrase using a more appropriate word(s).

It has been changed, please see line 159.

LINE 137-138: Rephrase the sentence to read like an action instead of a statement.

It has been changed, please see lines 160-161.

LINE 142: Corrosion rate does not have units of “mm/ a”.

It has been changed, please see line 165.

LINE 142: m and mt are not qualities of the metal sheet. Use appropriate scientific terma and avoid such ambiguity.

It has been changed, please see lines 165-166.

LINE 146: Rephrase “by three parallel specimens of each test”.

It has been changed, please see lines 168-169.

LINE 154: Write “evaluated by” instead of “performed with”.

It has been changed, please see line 177.

LINE 155-156: Write “determined” instead of “analyzed”.

It has been changed, please see line 179.

LINE 161: Write “ground” instead of “grounded”.

It has been changed, please see line 184.

LINE 161-169: How was the grinding operation carried out to get such surfaces? This must be clearly stated in the work.

Did you polishing in one direction? With only one Sic paper grade? In a rotating disc? Progressively with finer SiC papers until the target grade?

As you can see the lack of this information creates ambiguity in the mind of the reader and makes it difficult to appreciate the surface morphology data presented.

It has been changed, please see lines 118-129.

LINE 168: Write “gradually ground” instead of (grounded gradually”.

It has been changed, please see line 191.

LINE 180: Write “grinding” instead of “grounded”.

It has been changed, please see line 203.

LINE 184: Write “grinding” instead of “being grounded”.

It has been changed, please see line 207.

LINE 186: Write “PCO2” instead of “Pco2

It has been changed, please see line 209.

LINE 180-191: It is necessary to present data on both the surface profiles after 48 hours and the chemical composition of the surface after 48 hours to determine why the corrosion rates convergence after 48 hours occur irrespective of the initial surface roughness.

The SEM diagram in Figure 6 can also support the viewpoint of this article.

LINE 205: Write “shown” instead of “showed”.

It has been changed, please see line 229.

LINE 222: Write “the change in the” instead of “the change”.

It has been changed, please see line 245.

LINE 227-230: Write “day” or “days” as appropriate instead of “d”.

It has been changed, please see lines 250-253.

LINE 239: Delete “pitting”.

It has been changed, please see line 262.

LINE 244: Write “days” instead of “d”.

It has been changed, please see line 267.

LINE 247: Write “days” instead of “d”.

It has been changed, please see line 271.

LINE 251: Write “days” instead of “d”.

It has been changed, please see line 276.

LINE 254: Write “days” instead of “d”.

It has been changed, please see line 280.

LINE 257: Write “days” instead of “d”.

It has been changed, please see line 284.

LINE 265: Specify the alloying elements detected.

It has been changed, please see line 292.

LINE 266: Write “scale after” instead of “scale of”.

It has been changed, please see line 293.

LINE 267: “XRD spectra” is incorrect. Use appropriate description/words.

It has been changed, please see line 294.

LINE 271-273: Rephrase this sentence to improve clarity.

It has been changed, please see lines298-301.

LINE 276: “XRD spectra” is incorrect. Use appropriate description/words.

It has been changed, please see line 303.

LINE 278-279: What is the significance of this ratio “penetration rate/average corrosion rate ratio”? What does it measure or indicate? Highlight it in the work.

The“penetration rate/average corrosion rate ratio” is “pitting rate/average corrosion rate ratios”, which is used to characterize the pitting tendency.

LINE 284: Delete “was”.

It has been changed, please see line 311.

LINE 297-299: The meaning of this sentence is difficult to grasp. Rephrase to improve express your idea with clarity or delete the sentence.

It has been changed, the sentence was deleted。

LINE 320-321: What justifies these distinctions of E?

It has been changed, please see lines 344-345.

LINE 324-326: How was this measured? This was not indicated in the methods section.

It has been changed, please see lines 145-148.

LINE 328-336: This argument is unclear.

It has been changed, please see lines 352-360.

LINE 340: Sentence lacks clarity. Rephrase.

It has been changed, please see line 364.

LINE 340-342: Rephrase this sentence

It has been changed, please see lines 364-366.

LINE 346: Write “equations are as follows:” instead of “formula is as following:”.

It has been changed, please see line 370.

LINE 348: Write “form” instead of “formed”.

It has been changed, please see line 372.

LINE 360: “dependent” on what? The sentence is not clear.

It has been changed, please see lines 383-384.

LINE 365-366: How were the area ratios calculated? Again, this information is not presented in the materials and methods section.

It has been changed, please see lines 145-148.

LINE 374: How consistent are the profiles presented in Figure 14 across the sample surface to permit the calculation of surface areas?

The consistency of the profile of the sample surface shown in Figure 14 is sufficient for surface area calculations with an error of less than 5%.

LINE 392: Write “days” instead of “d”.

It has been changed, please see line 416.

LINE 405: Write “days” instead of “d”.

It has been changed, please see line 429.

LINE 409-415: This arrangement is not sufficiently supported by data presented. To talk of pitting and pitting potential as mentioned in Lines 412-415 electrochemical test data must be included that shows the pitting behaviour as the surface roughness changes. For this, a cyclic potentiodynamic polarization can be sufficient.

Thanks for your suggestion. Our laboratory does not currently have the conditions for high pressure electrochemical testing, and your suggestion will be our research direction.

LINE 439-441: Rephrase this sentence to improve clarity.

It has been changed, please see lines 459-474.

Reviewer 3 Report

The work studies the corrosion mechanism of J55 carbon steel in CO2 containing geothermal water. The topic is interesting, the methodology is clear and robust and the results are relevant. However, some modifications are needed.

Introduction

- I think that "pitting pits" is a tautology. Please use "pits" only.

Materials and methods

- What is "phi"? Make sure all symbols are explained in the manuscript.

- "x and y is the number of sampling points": this sentence may confuse the reader, please be more precise.

Results

- "The effect of surface roughness on the corrosion rate decreases with the increase of immersion time": the reductions are in absolute terms, but what about the percentage reduction? I mean, what is the ratio of the average corrosion rates for two Sa values, given the exposure time?

References

- The most recent references are 1 from 2022, 2 from 2021 and 1 from 2020. Please cite more recent references.

Author Response

An overview of the revision

Thanks for your suggestion and advice on our paper. We have revised the manuscript according to your detailed suggestions. We have carefully checked and improved the English writing in the revised manuscript. Words in red are the changes we have made in the text. We have revised the problems and discussed your arguments as follows.

Comment 1: Introduction

- I think that "pitting pits" is a tautology. Please use "pits" only.

Response 1: It has been changed, please see line 66 and line 68.

Comment 2: Materials and methods

- What is "ph"? Make sure all symbols are explained in the manuscript.

Response 2: pH value is a method of indicating the concentration of hydrogen ions. It is the negative logarithm of the concentration of hydrogen ions in an aqueous solution. pH value is a commonly used water quality indicator, which is generally not explained. All symbols were explained in the manuscript.

Comment 3: Materials and methods

- "x and y is the number of sampling Comments": this sentence may confuse the reader, please be more precise.

Response 3: It has been changed, please see lines 142-143.

Comment 4: Results

- "The effect of surface roughness on the corrosion rate decreases with the increase of immersion time": the reductions are in absolute terms, but what about the percentage reduction? I mean, what is the ratio of the average corrosion rates for two Sa values, given the exposure time?

Response 4: It has been changed, please see lines 459-462.

Comment 5: References

- The most recent references are 1 from 2022, 2 from 2021 and 1 from 2020. Please cite more recent references.

Response 5: References for recent years have been added.

Reviewer 4 Report

1. The abstract does not provide sufficient information for the article submission. The description was too brief and generic.

2. The author explains the fabrication technique in more detail and includes fabrication images for better understanding.

3. The author includes a recent literature survey on corrosion work and surface roughness.

4. The author explains more details in the conclusion section.

5. The results and discussion part could be improved. 

6. The author include novelty of your work.

7. The findings in the abstract and conclusion are expectable facts. You can give quantitative results that may be comparable with other materials.

8. What is the contribution of choosing CO2 containing geothermal water? 

Author Response

An overview of the revision

Thanks for your suggestion and advice on our paper. We have revised the manuscript according to your detailed suggestions. We have carefully checked and improved the English writing in the revised manuscript. Words in red are the changes we have made in the text. We have revised the problems and discussed your arguments as follows.

Comment 1: The abstract does not provide sufficient information for the article submission. The description was too brief and generic.

Response 1: It has been changed, please see lines 13-30.

Comment 2: The author explains the fabrication technique in more detail and includes fabrication images for better understanding.

Response 2: We have added to the text to make the experimental approach easier to understand, please see lines 122-127 and Figure 1.

Comment 3: The author includes a recent literature survey on corrosion work and surface roughness.

Response 3: It has been changed, please see lines 61-78.

Comment 4: The author explains more details in the conclusion section.

Response 4: We have updated the conclusions, please see lines 459-474.

Comment 5: The results and discussion part could be improved.

Response 5: It has been improved.

Comment 6: The author includes novelty of your work.

Response 6: It has been improved, please see lines 76-79.

Comment 7: The findings in the abstract and conclusion are expectable facts. You can give quantitative results that may be comparable with other materials.

Response 7: The current research mainly focuses on the impact of surface roughness on corrosion at the initial stage of corrosion, and does not evaluate the impact of metal surface roughness on the long-term corrosion. In particular, it lacks the effect of surface roughness on carbon steel corrosion in CO2 containing environments.

Comment 8: What is the contribution of choosing CO2 containing geothermal water?

Response 8: In recent years, geothermal energy has been used worldwide and has made positive contributions to carbon emission reduction. Geothermal energy with high CO2 content has also been gradually developed, but attention has not yet been paid to carbon steel corrosion in CO2 containing environments. This study can provide a reference for evaluating carbon steel corrosion in CO2 containing environments.

Round 2

Reviewer 2 Report

REVIEWER COMMENTS __ROUND__2__Effect of surface roughness on static corrosion behavior of J55 carbon steel in CO2 containing geothermal water at 65 ℃.

General Comments

The quality of the manuscript has been improved as the authors incorporated earlier comments.

A few errors persist and have been pointed out to the authors under specific comments below, along with suggestions o addressing the errors detected.

Some English Language editing would be helpful to improve fluidity.

Specific Comments

LINE 17: Delete £of the formation”.

LINE 24-25: The term “corrosion electromotive force” is strange and inappropriate in this context. Please remove its use in any part of the manuscript. You can use instead “the driving force for corrosion” etc.

LINE 26: Write “immersion” instead of “immerse”.

LINE 26: Write “continuous” instead of “complete”.

LINE 53: Write “could” instead of “would”.

LINE 59: Delete “recognition results”.

LINE 86: This phrase makes no sense; “The influence of temperature and pressure on reported that the corrosion ….”. Some words appear to be missing.

Rephrase to improve clarity.

LINE 86-88: If this sentence is based on previous work, make it clear and insert appropriate references.  Authors should note that any statement without citation is bound to be construed by a reader to be the authors’ thoughts and/or ideas.

LINE 113: Write “with a hole of bore diameter = 6mm drilled to enable suspension of samples in tests solutions resulting in an exposed area of 13.6 cm2” instead of  “bore diameter = 6mm with an exposed area of 13.6 cm2”.

LINE 158: Write “purge” instead of “purify”.

LINE 160: Delete “maintain”.

LINE 161: Write “conditions were maintained in “instead of “conditions in”.

LINE 165: Write “mass” instead of “quality”.

LINE 166: Write “mass” instead of “quality”.

LINE 174: Write “cross-section” instead of “cross-sectional”.

LINE 181: Write “an” instead of “the”.

LINE 203: Delete “being”.

LINE 265-285: In Figures 7,8,9, and 10 make it clear in the respective figure caption the significance of the big and small blue squares, respectively.   

LINE 300-301: The sentence should be rephrased to enhance clarity. It is difficult to understand n the present form.

LINE 311: Write “decreased” instead o “was decreased”.

LINE 354: Write “driving force” instead of “electromotive force”.

LINE 357: Write “rate” instead of “degree”.

LINE 364: Delete the sentence, or rephrase it to make sense. It is lacking clarity in the present form.

LINE 373: Write “impede” instead of “had a certain hindrance to”.

LINE 432-433: Write “driving force” instead of “electromotive force”.

LINE 464: Write “immersion” instead of “immerse”.

LINE 466: Write “immersion” instead of “immerse”.

LINE 471: Write ”At the initial stages of corrosion,” instead of “At the initial corrosion”.

Author Response

Response to Reviewer 2 Comments

An overview of the revision

Thanks for your suggestions and advices on our paper. We have revised the manuscript according to your detailed suggestions. Words in red are the changes we have made in the text. We have revised the problems and discussed your arguments as follows.

Comment 1: LINE 17: Delete £of the formation”.

Response 1: It has been changed, please see line 17.

Comment 2: LINE 24-25: The term “corrosion electromotive force” is strange and inappropriate in this context. Please remove its use in any part of the manuscript. You can use instead “the driving force for corrosion” etc.

Response 2: It has been changed, please see line 24 and lines 347-348 .

Comment 3: LINE 26: Write “immersion” instead of “immerse”.

Response 3: It has been changed, please see line 26.

Comment 4: LINE 26: Write “continuous” instead of “complete”.

Response 4: It has been changed, please see line 26.

Comment 5: LINE 53: Write “could” instead of “would”.

Response 5: It has been changed, please see line 53.

Comment 6: LINE 59: Delete “recognition results”.

Response 6: It has been changed, please see line 59.

Comment 7: LINE 86: This phrase makes no sense; “The influence of temperature and pressure on reported that the corrosion ….”. Some words appear to be missing.

Rephrase to improve clarity.

Response 7: The meaning of this sentence is repeated in the next sentence, and we have deleted it.

Comment 8: LINE 86-88: If this sentence is based on previous work, make it clear and insert appropriate references.  Authors should note that any statement without citation is bound to be construed by a reader to be the authors’ thoughts and/or ideas.

Response 8: It has been changed, please see line 88.

Comment 9: LINE 113: Write “with a hole of bore diameter = 6mm drilled to enable suspension of samples in tests solutions resulting in an exposed area of 13.6 cm2” instead of  “bore diameter = 6mm with an exposed area of 13.6 cm2”.

Response 9: It has been changed, please see lines 110-111.

Comment 10: LINE 158: Write “purge” instead of “purify”.

Response 10: It has been changed, please see line 156.

Comment 11: LINE 160: Delete “maintain”.

Response 11: It has been changed, please see line 158.

Comment 12: LINE 161: Write “conditions were maintained in “instead of “conditions in”.

Response 12: It has been changed, please see line 159.

Comment 13: LINE 165: Write “mass” instead of “quality”.

Response 13: It has been changed, please see line 163.

Comment 14: LINE 166: Write “mass” instead of “quality”.

Response 14: It has been changed, please see line 164.

Comment 15: LINE 174: Write “cross-section” instead of “cross-sectional”.

Response 15: It has been changed, please see line 172.

Comment 16: LINE 181: Write “an” instead of “the”.

Response 16: It has been changed, please see line 179.

Comment 17: LINE 203: Delete “being”.

Response 17: It has been changed, please see line 201.

Comment 18: LINE 265-285: In Figures 7,8,9, and 10 make it clear in the respective figure caption the significance of the big and small blue squares, respectively.   

Response 18: It has been changed, please see lines 284-288.

Comment 19: LINE 300-301: The sentence should be rephrased to enhance clarity. It is difficult to understand n the present form.

Response 19: It has been changed, please see lines 303-304.

Comment 20: LINE 311: Write “decreased” instead of “was decreased”.

Response 20: It has been changed, please see line 314.

Comment 21: LINE 354: Write “driving force” instead of “electromotive force”.

Response 21: It has been changed, please see line 357.

Comment 22: LINE 357: Write “rate” instead of “degree”.

Response 22: It has been changed, please see line 359.

Comment 23: LINE 364: Delete the sentence, or rephrase it to make sense. It is lacking clarity in the present form.

Response 23: It has been deleted.

Comment 24: LINE 373: Write “impede” instead of “had a certain hindrance to”.

Response 24: It has been changed, please see line 372..

Comment 25: LINE 432-433: Write “driving force” instead of “electromotive force”.

Response 25: It has been changed, please see lines 431-432.

Comment 26: LINE 464: Write “immersion” instead of “immerse”.

Response 26: It has been changed, please see lines 463-464.

Comment 27: LINE 466: Write “immersion” instead of “immerse”.

Response 27: It has been changed, please see line 465.

Comment 28: LINE 471: Write ”At the initial stages of corrosion,” instead of “At the initial corrosion”.

Response 28: It has been changed, please see line 470.
